# Spatial Distribution of Multi-Fractal Scaling Behaviours of Atmospheric XCO_2_ Concentration Time Series during 2010–2018 over China

**DOI:** 10.3390/e24060817

**Published:** 2022-06-11

**Authors:** Yiran Ma, Xinyi He, Rui Wu, Chenhua Shen

**Affiliations:** 1College of Geographical Science, Nanjing Normal University, Nanjing 210046, China; 201302054@njnu.edu.cn (Y.M.); 201345038@njnu.edu.cn (X.H.); 211302136@njnu.edu.cn (R.W.); 2Key Laboratory of Virtual Geographic Environment of Ministry of Education, Nanjing 210046, China; 3Jiangsu Center for Collaborative Innovation in Geographical Information Resource, Nanjing 210046, China

**Keywords:** spatial distribution, multi-fractal scaling behaviour, atmospheric XCO_2_ concentration, spatio-temporal thin plate spline interpolation approach, multi-fractal detrended fluctuation analysis

## Abstract

Exploring the spatial distribution of the multi-fractal scaling behaviours in atmospheric CO_2_ concentration time series is useful for understanding the dynamic mechanisms of carbon emission and absorption. In this work, we utilise a well-established multi-fractal detrended fluctuation analysis to examine the multi-fractal scaling behaviour of a column-averaged dry-air mole fraction of carbon dioxide (XCO_2_) concentration time series over China, and portray the spatial distribution of the multi-fractal scaling behaviour. As XCO_2_ data values from the Greenhouse Gases Observing Satellite (GOSAT) are insufficient, a spatio-temporal thin plate spline interpolation method is applied. The results show that XCO_2_ concentration records over almost all of China exhibit a multi-fractal nature. Two types of multi-fractal sources are detected. One is long-range correlations, and the other is both long-range correlations and a broad probability density function; these are mainly distributed in southern and northern China, respectively. The atmospheric temperature and carbon emission/absorption are two possible external factors influencing the multi-fractality of the atmospheric XCO_2_ concentration. **Highlight**: (1) An XCO_2_ concentration interpolation is conducted using a spatio-temporal thin plate spline method. (2) The spatial distribution of the multi-fractality of XCO_2_ concentration over China is shown. (3) Multi-fractal sources and two external factors affecting multi-fractality are analysed.

## 1. Introduction

In recent decades, fossil fuel combustion and industrial processes arising from the intensification of anthropogenic activities have caused significant increases in the global atmospheric carbon dioxide (CO_2_) concentration [1]. The excessively high atmospheric CO_2_ concentration has led to a series of ecological and environmental problems, such as global warming, rising sea levels, droughts and floods, degradation of the soil ecosystem, and reductions in grain production [2,3,4]. Under the background of climatic change and global governance challenges, CO_2_, as the most important and longest-lived greenhouse gas in the atmosphere, has attracted increasing attention from researchers and policymakers. Analyses of the spatial distributions of the multi-fractal scaling behaviours, including small or large fluctuations, multi-fractal strength and multi-fractal sources, of the atmospheric CO_2_ concentration records are helpful for understanding the dynamic mechanisms of CO_2_ concentration variations owing to carbon emission/absorption and climatic change.

In the past several years, an ever-growing number of studies have focused on the multi-fractal scaling behaviours of meteorological and air quality parameters in the field of atmospheric sciences using the well-established multi-fractal detrended fluctuation analysis (MFDFA) [5,6]. The MFDFA has been applied into time series concerning atmospheric temperature [7,8,9,10,11], precipitation [12,13], relative humidity [14], particulate matter concentrations [15], and many other measurements. A few investigations have focused on spatial distributions of the multi-fractal scaling behaviours for these meteorological elements [10,16,17]. These investigations, especially those concerning the spatial heterogeneity of meteorological elements, are more helpful for understanding the corresponding dynamic mechanisms.

However, to present, in the context of the analysis of the complexity and non-linearity of signals, the multi-fractal scaling behaviours of CO_2_ mainly concern two aspects: the carbon trade, and the atmospheric CO_2_ concentration record (with various time scales including hour, day, and month). In the literature, there are numerous investigations focusing on the scaling behaviours of the carbon trade [18,19,20,21,22,23,24]; however, relatively fewer investigations have focused on the atmospheric CO_2_ concentration [25,26,27,28], particularly from the perspective of spatial heterogeneity. Evidently, our knowledge of the scaling behaviours of atmospheric CO_2_ concentrations is limited. For instance, some researchers believed that atmospheric CO_2_ concentration records were characteristic of a long-term memory and obeyed a power law [25]; in contrast, others argued that thermospheric CO_2_ power did not exhibit power-law behaviours [28]. Their standpoints contradict each other.

As the direct sampling of greenhouse gas concentrations, especially in the upper atmosphere, requires considerable effort and massive costs [29], current ground-based measurement stations are sparse and uneven [29]. Spatio-temporal CO_2_ concentration records are not plentiful. Our knowledge of the scaling behaviours of CO_2_ concentration is further limited by the insufficiency of spatio-temporal CO_2_ concentration records.

Nonetheless, Japan launched the world’s first greenhouse gas observation satellite, the Greenhouse Gases Observing Satellite (GOSAT) in 2009 [30,31]. This satellite has offered a potential way to complement existing measurements of CO_2_ concentrations through space-based remote sensing. Since CO_2_ concentration values from GOSAT have been released, they have been practically applied in many scientific investigation works [32,33,34,35,36,37]. Although space-based observations can obtain CO_2_ concentrations for global coverage and have a high measurement density, these satellite observations have gaps and are irregularly positioned owing to certain limitations including clouds, the observational mode of the satellite [30], and inversion algorithms. It is necessary to fill these gaps, and geospatial statistics might be an effective tool for this task [38]. Ref. [39] incorporated temporal variability to improve a Kriging geo-statistical analysis [40] of satellite-observed CO_2_ concentration in China, and presented a reliable interpolation of the XCO_2_ concentration. However, in the spatio-temporal Kriging interpolation [40], it was very difficult to select a reliable and reasonable semi-variance model and its parameter [40]. Inappropriate selections of models and parameters can lead to significantly improper results and low efficiencies in interpolation.

In this study, to overcome the lack of knowledge regarding the multi-fractal scaling behaviour, we investigate the spatial distribution of the multi-fractal scaling behaviour of an atmospheric XCO_2_ concentration time series over China from 2010−2018, and get an insight into the dynamic mechanisms of CO_2_ concentration changes. China covers a large geographical area, with wide climatic variety, complicate topography and plentiful vegetation coverage types. Within this context, it would be interesting and ideal to examine the spatial distribution of the multi-fractal properties over the whole of China. We propose an improved spatio-temporal interpolation approach to implement spatio-temporal interpolation of atmospheric XCO_2_ concentration, and use MFDFA to estimate their multi-fractal scaling exponents. The results of this study are respected to contribute to the knowledge regarding the multi-fractal scaling behaviours, their spatial distribution, multi-fractal sources, and possible external influencing forces.

## 2. Study Area and Data Sources

### 2.1. Study Area

The study region covers the Chinese mainland with latitude from 21° to 54° N and longitude from 70° to 138° E, divided into a total of 4189 grid points with a spatial resolution of 0.5° × 0.5°. Seas and islands in China are excluded, as the greenhouse gas tendency largely varies between land and sea. Taiwan, Hong Kong, and Macao are also excluded, owing to an absence of data sources.

Figure 1 shows spatial pattern of major vegetation types (http://data.tpdc.ac.cn/zh-hans/data/eac4f2cf-d527-4140-a35d-79992957f043/ (accessed on 10 June 2022)). The major land-cover types in China are forest, grassland, paddy fields and rain-fed croplands, which are mainly located in northeast China, northern China, and southern China, respectively. More specifically, a deciduous broadleaf forest is located in northeast China, and an evergreen broadleaf forest is distributed in southern China. Most of the land-cover types in the earth’s terrestrial ecosystems are found in China [41].

There are four typical grid points with different climate, land cover and vegetation types, marked by red coloured circles (A, B, C and D). The grid point A (109°15′ E, 42°15′ N) is located in Inner Mongolia, with low air temperature and little precipitation, where land is mainly covered by grassland or barren land like desert. The grid point B (116°45′ E, 23°15′ N) is placed in Guandong province, with high air temperature and plentiful precipitation, where major land cover is forestry. The grid point C (120°45′ E, 32°45′ N) is situated in Jiangsu province, with median air temperature and moderate precipitation. The double crop system is the main characteristic of this region. The grid point D (100°54′ E, 36°17′ N, 3816-m altitude) is WLG (ground observational station), with low air temperature and high altitude. WLG is the highest atmospheric background reference monitoring station in the world. As WLG station is far away from the built-up area, the CO_2_ concentration measured by an instrument is highly accurate.

### 2.2. Data Collection and Pre-Processing

#### 2.2.1. Column-Averaged Dry-Air Mole Fraction of Carbon Dioxide (XCO_2_) from GOSAT

The Greenhouse Gases Observing Satellite was launched in 2009 by Japan, and circles around the earth in a sun-synchronous orbit at a 666-km altitude with the descending node at approximately 12:48 local time and a 3-day recurrence [30,31]. GOSAT was carefully and specifically designed to record two concentrations of greenhouse gases from space: the column-averaged dry-air mole fraction of carbon dioxide (XCO_2_) and methane (XCH_4_), respectively [30]. The GOSAT Level 2 data are the CO_2_ dry air mixing ratios (XCO_2_), defined by the ratio of the total number of CO_2_ molecules to that of dry molecules, not only near the earth surface, but also in the entire vertical column to the top of the atmosphere [30].

We acquired and rearranged the monthly XCO_2_ data from GOSAT from January 2010 to December 2018. A total of 10367 observations were available. Theoretically, a total of 452,412 (4189 grid points × 9 years × 12 months per year) observations should be available for all of the grid-points in the study region. Clearly, various observations in the spatial and temporal dimensions were missing. For instance, all of the CO_2_ concentration values in January 2015 were absent. In an additional case, the CO_2_ concentration values in May 2010 were available, but the ones in July 2010 were missing at the same location. Therefore, a spatio-temporal thin plate spline interpolation approach (STTPS) needed to be developed.

#### 2.2.2. In-Situ CO_2_ Concentration Values from Ground-Based Station

To examine the interpolation accuracy of XCO_2_ concentration, the in-situ CO_2_ concentration values were collected from the Waliguan meteorological ground-based observation station. The atmospheric environment and components of the WLG station were very stable, owing to its distance from built-up areas. Therefore, the precision and reliability of the in-situ observed values of the CO_2_ concentration at WLG station are high. Notably, the CO_2_ concentration values for April 2011, November 2011, December 2011, and August 2014 were absent from the WLG during 2010−2018.

## 3. Methodology

### 3.1. Spatio-Temporal Thin Plate Spline Interpolation

As the number of observations of spatio-temporal XCO_2_ concentration is finite, it is difficult to use these XCO_2_ concentration observations to realize the valuable analysis. The spatio-temporal interpolation, thus, is a simple and economical approach for obtaining sufficient observations of the spatio-temporal XCO_2_ concentration.

Let us suppose that in the study region, there is a set of CO_2_ data values in space-based grid points. To obtain a continuous surface consisting of uniformly distributed grid points with a spatial resolution of 0.5° × 0.5°, the interpolation is conducted based on a radial basis function (RBF) [42,43]. The RBF interpolation (a distance-based function) is written as Equation (1), as follows:(1)$$\varphi \left(x,y\right)=\sum_{j=1}^{n}A_{j}d_{j}^{2}\log d_{j}+b_{1}+b_{2}x+b_{3}y$$

In the above, *d_j_* is a Euclidean distance between an interpolated point and the *j*th known point, and respective *x* and *y* are the latitude and longitude coordinates of the interpolated point. Among the distance-based functions, the thin plate spline function is the most fundamental type of interpolation one. It is considered as a bending thin plate, and can pass through all known points with minimal bending energy. Hence, in Equation (1), there are a total of *n* + 3 interpolation coefficients, namely, **A**_j_ (*j* = 1, 2, 3…, *n*), *b*_1,_
*b*_2_, and *b*_3_ that need to be estimated. This is called as the spatial thin plate spline interpolation method (abbreviated as STPS).

To overcome singularity when estimating the interpolation coefficients, three additional constraint conditions derived from the spline function properties should be incorporated as follows:(2)$$\sum_{j=1}^{n}A_{j}=0,\sum_{j=1}^{n}A_{j}x_{j}=0,\;\sum_{j=1}^{n}A_{j}y_{j}=0$$

When Equations (1) and (2) are combined, all the interpolation coefficients can be estimated easily. Next, temporal variability is incorporated to improve STPS interpolation of the satellite-observed CO_2_ measurements. Equation (1) is thus rewritten into Equation (3) as follows:(3)$$\varphi \left(x,y,t\right)=\sum_{j=1}^{n}A_{j}d_{j}^{2}\log d_{j}+b_{1}+b_{2}x+b_{3}y+b_{4}\times \tau \times t$$

Here, φ(x,y,t) is an interpolated CO_2_ concentration, *t* is the time, and $d_{j}^{2}\log d_{j}^{}$ is a RBF with the minimum spatio-temporal surface curvature. *d*_j_ is a spatio-temporal distance between the interpolated point and the *j*th known point. It is denoted by $d_{j}^{2}=d_{sj}^{2}+\tau^{2}\times d_{tj}^{2}$, where $d_{sj}^{}$ is a Euclidean distance in space, τ × *d*_tj_ is one in time,$d_{\mathrm{tj}}=t-t_{j}$, and *τ* is a spatio-temporal ratio. There are the interpolation coefficients of *n* + 4 (A_j_ (*j* = 1, 2, 3…, *n*), *b*_1_, *b*_2_, *b*_3_, and *b*_4_). Four additional constraint conditions analogous to those in Equation (2) need to be incorporated before estimating the interpolation coefficients. This is called as the spatio-temporal thin plate spline interpolation method.

The optimal *τ*, denoted as τ*, is determined by experiments as follows. First, 90% of the known data points are randomly chosen from the XCO_2_ concentration dataset as the interpolation control points for estimating the interpolation coefficients, and the remaining 10% data points are used to examine interpolation accuracy. Second, a different value of *τ* is set to get a different interpolation variance. A plot of the interpolation variance against *τ* is drawn. When the interpolation variance is the smallest, the corresponding *τ* is then set as τ*. To obtain τ* as fast as possible, the dichotomy is generally adopted.

### 3.2. Multi-Fractal Detrended Fluctuation Analysis

The generalised Hurst exponent (scaling exponent) is important for revealing the long-term correlation in time series [44,45]. A detrended fluctuation analysis is a powerful and effective tool for estimating the generalised Hurst exponent of non-stationary time series. Here, we briefly recall the detrended fluctuation analysis (DFA) [44] and MFDFA [5].

Assume that there is one time series {*x*(*k*)}, *k =* 1, 2,…, *N*, where *N* is the length of this series. First, one must calculate the profile, as follows:(4)$$Z(i)=\sum_{k=1}^{i}(x(k)-\bar{x}),i=1,2,3,\ldots ,N$$

In the above, $\bar{x}$ denotes the averaging over the whole time series {*x*(*k*)}. Next, one needs to divide the profile *Z*(*i*) into *N_s_ =* int (*N*/*s*) non-overlapping segments of equal length *s*. As the series’ length *N* is not always a multiple of the given time scale *s*, a short part may remain at the end of each profile. To avoid discarding this part of the series, the same procedure is repeated starting from the opposite end of each profile, and 2*N_s_* segments are thus obtained together. The detrended covariance is determined as follows:(5)$$F_{}^{2}(s,\lambda )=\frac{1}{s}\sum_{j=1}^{j=s}{\left[Z((\lambda -1)s+j)-z_{\lambda }(j)\right]}^{2}$$

Here, *z**_λ_*(*j*) is the fitting polynomial in box *λ*. Next, the fluctuation function is defined in Equation (6)
(6)$$F(s)={\left\{\frac{1}{2N_{s}}\sum_{\lambda =1}^{2Ns}\left[F^{2}(s,\lambda )\right]\right\}}^{\frac{1}{2}}$$

Actually, F(*s*) will increase with the time scale *s*. The linear relationship on a log-log plot exhibits the existence of a power law F(*s*) ∝ *s*^H^. The scaling exponent H, called the generalized Hurst exponent, represents the correlation degree. If H = 0.5, then the signal is random and uncorrelated, i.e., a white noise; if H > 0.5, the signal is persistently auto-correlated; if H < 0.5, the signal is anti-persistently auto-correlated.

Kantelhardt et al. developed the MFDFA [5] based on DFA [44]. Equation (6) can be rewritten into Equation (7), as follows:(7)$$F_{q}^{}(s)={\left\{\frac{1}{2N_{s}}\sum_{\lambda =1}^{2Ns}\left[F^{2}{\left(s,\lambda \right)}^{\frac{q}{2}}\right]\right\}}^{\frac{1}{q}}$$

The varying moment *q* generally takes any real value. Similarly to the above, the linear relationship on a log-log plot exhibits the existence of a power-law F*_q_*(*s*) ∝ *s^h^*^(*q*)^. If the time series is mono-fractal, *h*(*q*) is independent of *q*; in contrast, if the time series is multi-fractal, *h*(*q*) depends on *q*. For *q* > *0*, *h*(*q*) describes the behaviour of the boxes with large fluctuations, whereas for *q* < 0, *h*(*q*) depicts the behaviour of the boxes with small fluctuations. Via a Legendre transform, we can obtain singularity strength and fractal dimension in Equations (8) and (9), as follows:(8)$$\{\begin{array}{c} \alpha =\frac{d\tau }{dq}=h(q)+q\frac{dh(q)}{dq}\\ \Delta \alpha =\alpha_{\max}-\alpha_{\min} \end{array}$$
(9)$$\{\begin{array}{c} f(\alpha )=\alpha q-\tau (q)=q(\alpha -h(q))+1\\ \Delta f(\alpha )=f(\alpha_{\min})-f(\alpha_{\max}) \end{array}$$

In the above, τ(*q*) = *qh* (*q*) −1, the quantity *α* is singularity strength (or Hölder exponent), whereas *f*(*α*) is a singularity spectrum for expressing the dimensionality of the subset of the time series [5]. A large Δα indicates that the multi-fractality is strong, whereas a small Δα suggests that the multi-fractality is weak. There are two types of multi-fractal sources: (i) a broad probability density function, and (ii) long-range correlations [5]. To distinguish multi-fractality sources, we might use a shuffle series and a surrogate series (or phase-randomised series). If the shuffle series has any multi-fractality, it would be owing to the broad probability density function; whereas if the surrogate series exhibits multi-fractality, it would be owing to the long-range correlations. In nature, the multi-fractality is sometimes driven by one of the sources or by both simultaneously.

## 4. Results

### 4.1. Accuracy Evaluation of the Interpolated Monthly XCO_2_ Concentration

The STTPS is applied to interpolate the monthly XCO_2_ concentrations from January 2010 to December 2018 over China. To examine the interpolation accuracy of XCO_2_ concentration, the average over the square of difference between the interpolated value and observed value is denoted by σ^2^. The smaller the value of σ^2^, the higher the interpolation accuracy is.

The two curved lines of the in-situ observed values and interpolated values at WLG (the ground-based station) during the identical period are shown in Figure 2a, where the black solid curved line is the observed in-situ value, and the dotted line is the interpolated one. Their σ^2^ is 7.88. As a whole, the trends of the two curved lines are in basic agreement, reflecting the identical seasonal fluctuations and inter-annual growth of the XCO_2_ concentration.

To confirm that STTPS has the advantage over STPS interpolation, the STPS interpolation method is also applied for the same data sources. Figure 2b exhibits the comparison between the observed values and interpolated values at WLG. Their σ^2^ is 9.95. Apparently, the value of σ^2^ when using STTPS is lower than that when using STPS approach. Furthermore, the peaks in the interpolated values when using STPS approach and those in the observed values do not appear simultaneously in Figure 2b, whereas the peaks in the interpolated values using STTPS and those in the observed values occur almost simultaneously in Figure 2a. Therefore, the interpolated values using STTPS have rather high interpolation accuracy.

To further evaluate the interpolation accuracy, a ten-fold cross validation method was employed. In particular, 90% of the satellite-based CO_2_ data points were randomly selected as interpolation control points, and the remaining 10% were used as interpolation validation points. After the optimal τ* was determined, STTPS was conducted. The identical interpolation process was repeated 100 times, and the average values of the evaluation indicators, such as the mean absolute error (MAE), mean square error (MSE), and root mean square interpolation error (RMSE), were calculated. If the MAE, MSE and RMSE are close to zero, it means that the interpolated values are approximate to the observed values, i.e., the interpolation accuracy is high. Table 1 shows the respective averages of MAE, MSE, and RMSE using STTPS. Our results were compared with those in Ref. [39]. It is intuitively inferred that interpolation values are consistent from the perspective of a thematic map, and that the values σ^2^ are roughly equivalent to each other. The interpolation accuracy is good.

### 4.2. Spatial Distribution of Multi-Fractal Scaling Behaviour

#### 4.2.1. Atmospheric XCO_2_ Multi-Fractality of Four Typical Grid Points

There are a total of 4189 grid points in the study region (See Figure 1). We analysed the multi-fractal behaviour of the month-average XCO_2_ concentration time series for every grid point. For convenience of discussion, we first analyzed multi-fractal behaviours of four typical grid points.

The temporal evolution and profile of the original month-average of XCO_2_ concentration time series at the WLG ground observational station from 2010 to 2018 are shown in Figure 2a. Apparently, the trends and approximate year-periodical fluctuations are easily observed. To reduce the multi-fractality uncertainties caused by the year-periodical fluctuations, the global year-periodical fluctuation should be removed by using a fitted model, as follows:(10)$$x(t)=\beta t+a_{1}\sin(\frac{2\pi }{period_{1}}t+\varphi_{1})+a_{2}\sin(\frac{2\pi }{period_{2}}t+\varphi_{2})+a_{3}\sin(\frac{2\pi }{period_{3}}t+\varphi_{3})+C+res(t)$$

In the above, *x*(*t*) is a month-average XCO_2_ concentration, *β* is a slope, *C* is an intercept, *res*(*t*) is a remainder component, and *period*_1_–*period*_3_ are the approximate annual period, half-annual period, and seasonal period, respectively. Ref. [7] recommended using a seasonal-trend decomposition procedure based on Loess (STL) to decompose a time series [46]. We compared the periodicity strength of the remainder component in Equation (10) with that from STL method using the wavelet analytical technology. As a result, the periodical strength of the remainder component in Equation (10) is too small to be seen. The fitted model in Equation (10) is more appropriate than the STL method. DFA3 is recommended to estimate generalised Hurst exponent *h*(*q*) in MFDFA.

Figure 3 shows the generalised Hurst exponents of the original series, shuffle series, and surrogate series against *q* for four typical grid points A, B, C, and D, respectively. Herein, the panels (A1−D1) express respective *h*(*q*) of four original series with *q*, panels (A2−D2) are respective *h^suf^*(*q*) of four shuffle series against *q*, and panels (A3−D3) are respective *h^sur^*(*q*) of four surrogate series versus *q*. The *q* ranges from −0.5 to 2.5, as the original series’ length is somewhat short. As seen, the exponents *h*(*q*) of the original series generally decrease with the increasing *q*. The respective value Δ*h* = max(*h*(*q*)) − min(*h*(*q*)) in the panel (A1 and D1) is relatively small, indicating that their multi-fractality is weak in the grid point A and D; whereas the respective value Δ*h* in panel (B1 and C1) is comparatively large, revealing that their multi-fractality is strong in the grid point B and C.

Figure 4 shows singular spectra of four typical grid points A, B, C, and D, respectively. The spectra width ∆α exhibits multi-fractal strength. The panels (b and c) show a strong multi-fractality, whereas the panels (a and d) display a weak multi-fractality. This finding is in agreement with Figure 3. In addition, the two singular spectra in panels (b and c) are asymmetrical, indicating that their multi-fractality dominantly stem from a large fluctuation of the original series.

To distinguish between the multi-fractal sources for the above four typical grid points, we first shuffled the original series and then estimated h^suf^(*q*) with *q* from −0.5 to 2.5. Next, we repeated the above procedure 1000 times, and ultimately averaged h^suf^(*q*) and calculated variance of h^suf^(*q*)(denoted as σ_h_(*q*)). Consequently, the maximum variance σ_h_(*q*) for the grid points (A and D) is 0.15. The panels (A2–D2) in Figure 3 show the averaged h^suf^(*q*) versus *q*. Δh^suf^ = max(h^suf^(*q*)) − min(h^suf^(*q*)) is defined. We judged that if Δh^suf^ < 0.15, the shuffle series is mono-fractal, indicating that its multi-fractality is owing to long-range correlations; otherwise the multi-fractal source is a broad probability density function. This criterion is a bit crude, as the original series length is finite.

Subsequently, similarly to the shuffle procedure, we repeatedly generated the surrogate series 1000 times by using phase-randomization, then estimated and ultimately averaged h^sur^(*q*). The panels (A3–D3) in Figure 3 show the averaged h^sur^(*q*) versus *q*. We judged that if Δh^sur^ < 0.15, the surrogate series is mono-fractal, and the multi-fractal source is a broad probability density function; otherwise, the multi-fractal source is long-range correlations.

As shown in the panels B2, B3, C2 and C3 in Figure 3, we discerned that the multi-fractal source originates from long-range correlations, and not from a broad probability density function. We determine whether the remainder components of the two original series in grid points B and C are characteristic of a long-tail distribution using the Statistical Package for Social Sciences. As a result, the two remainder components do not pass the statistical test for the long-tail distribution under a confidence interval of 95%.

#### 4.2.2. Spatial Distribution of Multi-Fractal Scaling Behaviour

The spatial distribution of the scaling exponents H = h(q = 2) for all of the original series is shown in Figure 5a. The values of h(q = 2) are greater than 0.6 in most regions in China, indicating that XCO_2_ concentration records in most regions show a persistent auto-correlation. This finding suggests that the XCO_2_-concentration fluctuation continues to increase from 2010 to 2018. However, in a few regions like the central part of Shanxi province, the values of h(q = 2) range from 0.44 to 0.55, indicating that the XCO_2_-concentration series are uncorrelated.

Figure 5b shows the spatial distribution of multi-fractal strength Δα of the original series. When Δα < 0.3, the original series are mono-fractal otherwise are multi-fractal [10]. As a whole, the multi-fractal behaviour in western China is stronger than that in eastern China. Within eastern China, the multi-fractal behaviour in the southern part is stronger than that in northern part, whereas in the upstream areas between the Yangtze River and Yellow River and along the Yangtze River Basin, the multi-fractal behaviour is weak. We notice that there are few grid points with Δα ≈ 0, i.e., less than 1% of the total grid points.

Figure 5c portrays the spatial distribution of the values α_max_ of the original series, disclosing the spatial distribution of the small fluctuations. Apparently, the spatial distribution in eastern and western China somewhat resembles that of Δα. The small fluctuation in western China is stronger than that in eastern China. Within eastern China, the small fluctuation in the south region is stronger than that in north region.

Figure 5d displays the spatial distribution of the values α_min_ of the original series, reflecting the spatial distribution of the large fluctuations. The grid points with the larger α_min_ values are generally scattered throughout the whole of China. The spatial distribution does not have evident regularity. It likely reflects the original series’ complexity depending on climate change and spatial distribution of vegetation coverage.

The spatial distribution of the multi-fractal sources is shown in Figure 6. There are mainly two types of multi-fractal sources: one is long-range correlations, and the other is both long correlations and a broad probability density function. The former is mainly distributed in southern, middle and northeast China, and accounts for 51% of the grid points. The approximately triangle-shaped pattern is shown in southern and middle China (light blue region in Figure 6). The latter is located in northern and middle China with an inverted-triangle-shaped pattern (yellow region part in Figure 6), and accounts for the remaining 44% of all of the grid points.

In summary, the scaling behaviours are a positive and persistent auto-correlation in most regions. The multi-fractal strength of the original XCO_2_ concentration series is different, as it is strong in western China and weak in eastern China. There are two multi-fractal sources: one is long-range correlations and the other is both long-range correlations and a broad probability density function [5].

## 5. Discussions

The multi-fractal strength in China is different, i.e., strong in western China and weak in eastern China. The CO_2_ concentration in the atmosphere is usually controlled by two physical mechanisms: climatic changes like atmospheric temperature, and carbon emission/absorption. Owing to two physical mechanisms, the dynamic mechanisms of the CO_2_ concentration seem more complicated than a single physical mechanism.

Refs. [9,10] reported that the strong multi-fractality of atmospheric temperature was mainly located in southern and northern China, while the weak multi-fractality was along the Yangtze River Basin. By comparison, the spatial distribution of the multi-fractality of XCO_2_ concentration in Figure 5b somewhat resembles that of the atmospheric temperature [9,10]. This finding is easily understood. An increase in atmospheric temperature will cause CO_2_ molecules to leave open systems out. The increasing atmospheric temperature can thereby lower the CO_2_ concentration. We can, hence, preliminarily infer that the atmospheric temperature, as an external force, has an effect on the multi-fractality of the XCO_2_ concentration, i.e., has an influence on small fluctuations and large fluctuations of XCO_2_ concentration.

Figure 5c shows that the small fluctuations of XCO_2_ concentration in southern China are weaker than in northern China. We speculated that vegetation absorption is one of external forces causing the small fluctuations of XCO_2_ concentration. The absorption magnitude and absorption time length have strong influences on small fluctuations. We noticed that vegetation coverage in eastern China is high, whereas it is very low in western China owing to the desert. Although most south areas of the Yangtze River and the northeast region (eastern region in Inner Monglia, Heilongjiang and Jilin) in China are covered by forestry, growing season of forestry is different. The growing season in south areas of the Yangtze River is usually from April to October even November in some places, but in northeast region, is from May to September. Clearly, the length of absorption time of CO_2_ in south areas of the Yangtze River is longer than northeast region. The number of small fluctuations in the original series for XCO_2_ concentration in south areas of the Yangtze River thereby is more than northeast region, indicating that the values α_max_ in south areas of the Yangtze River are greater than northeast region. Meanwhile, in the upstream areas between the Yangtze River and Yellow River and along the Yangtze River Basin, double cropping system has influences on fluctuations of CO_2_ concentration to some extent. The wheat harvest is frequently completed at the end of May every year in this region, and then corn or soybeans are planted immediately in early June. Two absorption peaks of CO_2_ concentration often appearing in May and July, respectively, are clearly discerned. Hence, compared with south areas of the Yangtze River, the number of small fluctuations is less, and the values α_max_ is small.

The spatial differentiation of the vegetation cover has a strong effect on CO_2_ concentration. Besides that, human-induced disturbances including industry production, vehicle emissions, and nature-induced variations like local forest fires, biomass burning, and cultivated land destruction, will affect the variation in CO_2_ concentration owing to emissions. Thereby, we speculate that it is possible that two external forces affect the multi-fractality of the CO_2_ concentration: one is climatic change like atmospheric temperature, and the other is carbon emission/absorption. However, the specific physical mechanisms as to how these two external forces affect the multi-fractal sources of the CO_2_ concentration merit further investigation.

## 6. Conclusions

In this work, we investigated the spatial distributions of multi-fractal scaling behaviours of an interpolated XCO_2_ concentration time series from 2010–2018 at a total of 4189 grid points by applying MFDFA method. The main results can be summarised as follows.

(I)We improved a spatio-temporal thin plate spline interpolation approach, and conducted interpolation of the monthly XCO2 concentrations over China from 2010−2018 based on GOSAT observations of XCO2. The interpolation accuracy of spatio-temporal thin plate spline interpolation approach was higher than a spatial thin plate spline interpolation one. The interpolated XCO2 concentration is highly accurate and is useful in analyzing multi-fractal scaling behaviours.(II)We found that the scaling behaviours of XCO2 concentration show a positive and persistent auto-correlation in most regions. The scaling behaviours of CO2 did not always obey power laws.(III)The multi-fractal strength of XCO2 concentration is different, i.e., strong in western China and weak in eastern China. There are two types of multi-fractal sources: one is long-range correlations, and the other is both long-range correlations and a broad probability density function. Two types are mainly distributed in southern and middle China with a triangle-shaped pattern and in northern China with an inverted-triangle-shaped pattern, respectively. Two external forces are likely to have influences on multi-fractality: the climatic change like atmospheric temperature and the carbon emission/absorption.

There is a limitation in this study, that is, the original XCO_2_ concentration time series length is relatively short. Because the spatial and temporal resolutions of the XCO_2_ concentration are low, the data sources directly available in the investigation are insufficient. If we use fewer observational data sources to conduct the spatio-temporal interpolation, the uncertainty in the interpolated data would greatly increase. Thus, mining suitable data sources of long CO_2_ concentration series with high quality (direct observational values instead of interpolation ones) is another important task in the future.

## Figures and Tables

**Figure 1 entropy-24-00817-f001:**
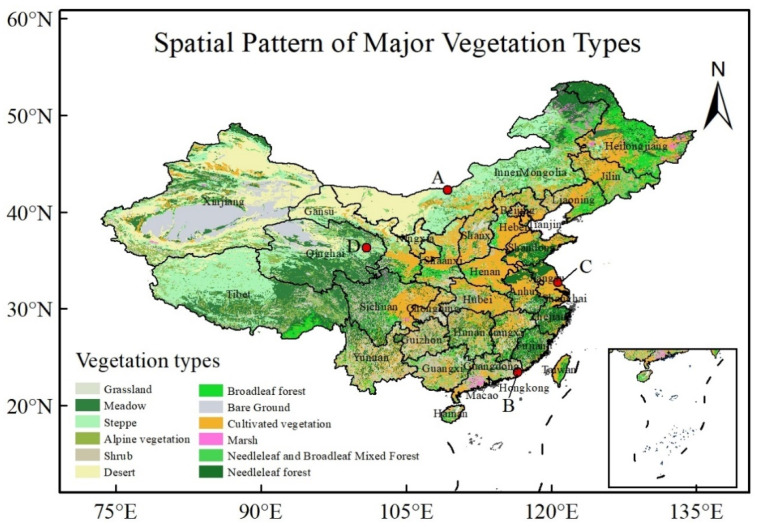
Spatial pattern of major vegetation types and four typical grid points with different climate, land cover and vegetation.

**Figure 2 entropy-24-00817-f002:**
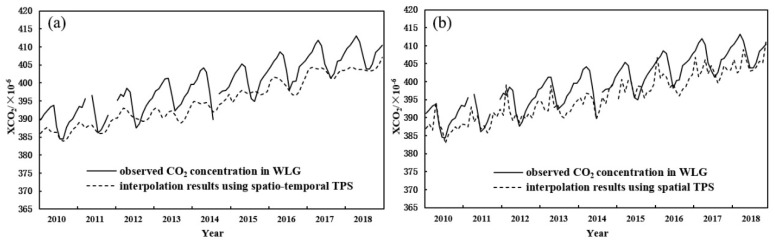
Comparison between the observed in-situ values and interpolated ones in WLG. (**a**) for STTPS (spatio-temporal TPS) and (**b**) for STPS (spatial TPS).

**Figure 3 entropy-24-00817-f003:**
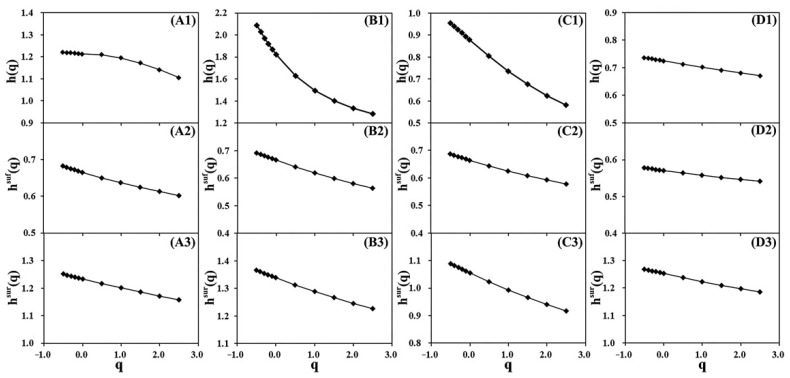
Generalised Hurst exponents of the original series, shuffle series, and surrogate series against *q* for four typical grid points A, B, C, and D, respectively. Panels (**A1**–**A3**) for grid point A, panels (**B1**–**B3**) for grid point B, and so on.

**Figure 4 entropy-24-00817-f004:**
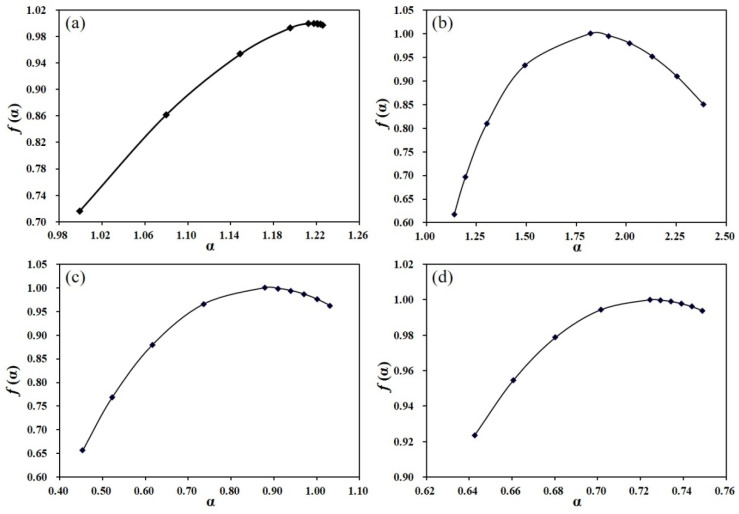
Singular spectra of four original series for grid points a, b, c, and d. Panel (**a**) for A, panel (**b**) for B, and so on.

**Figure 5 entropy-24-00817-f005:**
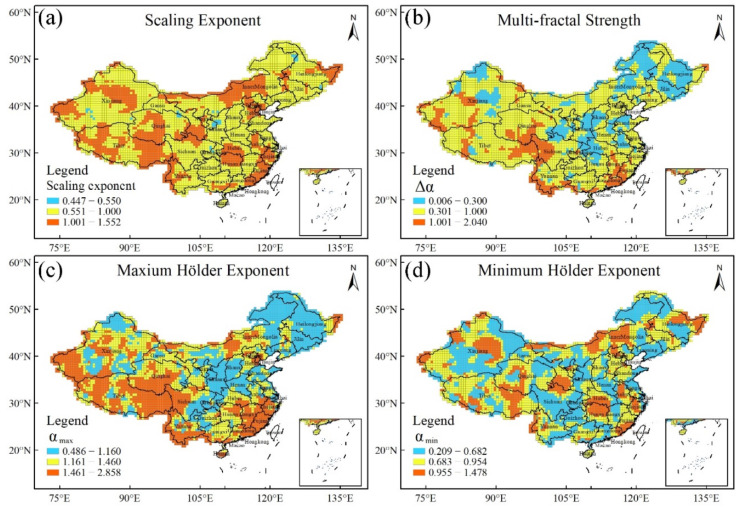
Spatial distribution of multi-fractal scaling behaviours for the original series. Panel (**a**) for scaling exponents, panel (**b**) for multi-fractal strength, panel (**c**) for maxium Hölder exponent and panel (**d**) for minimum Hölder exponent.

**Figure 6 entropy-24-00817-f006:**
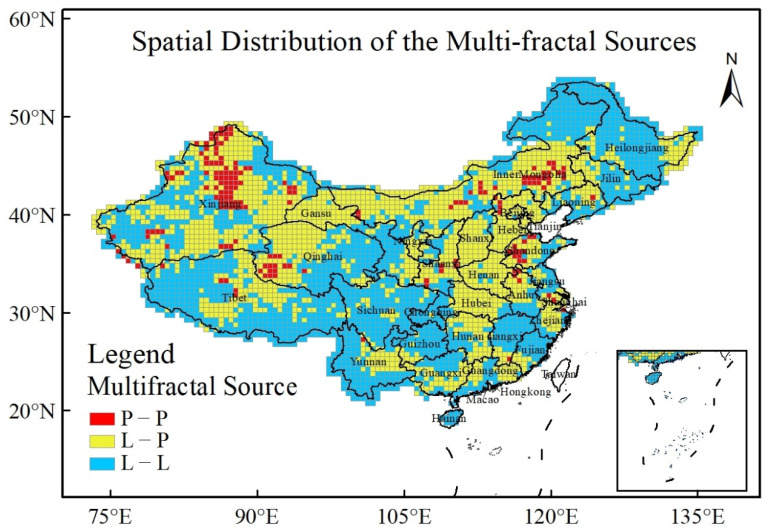
Spatial distribution of the multi-fractal sources. P-P is a broad probability density function, L-P is both long-range correlations and a broad probability density function, and L-L is long-range correlations.

**Table 1 entropy-24-00817-t001:** Variances of the STTPS interpolation.

Year	MAE	MSE	RMSE
2010	1.61	4.43	2.11
2011	1.59	4.99	2.23
2012	1.40	3.47	1.86
2013	1.46	4.24	2.06
2014	1.49	4.43	2.10
2015	1.39	3.37	1.84
2016	1.52	4.77	2.19
2017	1.38	3.80	1.95
2018	1.51	4.14	2.03
average	1.48	4.18	2.04

## Data Availability

The datasets used or analysed during the current study are available from the corresponding author on reasonable request.

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
