# Peer review of "Spatial Distribution of Multi-Fractal Scaling Behaviours of Atmospheric XCO2 Concentration Time Series during 2010–2018 over China"

_entropy, 2022, doi:10.3390/e24060817_

Round 1
Reviewer 1 Report
The manuscript reports a study exploring the spatial scaling behaviors of space-based CO2 measurements and their spatial distribution in the regions of China. A multi-fractal detrended fluctuation analysis (MDFA) was applied in the study. The data were from the GOSAT satellite, and missing data were interpolated using a spine interpolation method. The study confirmed the XCO2’s multi-fractal behaviors and discussed influencing factors. It is interesting to see how different the behaviors are between the regions of China.
Overall, the manuscript is well-developed, as I can follow the narratives and complex formulas with my limited knowledge of MEFA. It will be helpful if terms like “long-range correction” and “a broad density function” can be explained with physical meanings and discuss what they imply to CO2 emission and ecosystem CO2 uptake behaviors in each region?
Just a few places need revisiting.
Line 7-8, the first sentence of the Abstract sounds unnecessary.
Line 91, delete “Fig. 1 shows”
Line 101, the figure caption needs to offer more information about each study area so that the readers can know about the land cover/vegetation/climate A, B, C, and D without referring back to the text. Also, in the text, explain why or how to choose these four locations?
Please apply this comment to other figures’ captions.
Line 128-208, double-check all equations and related variable symbols and explanations
Line 145, any citation for the STPS method?
Line 340-341, provide more explanations of “long-range correction” and “a broad density function” before these terms are used.
Author Response
Open Review#1
Comments and Suggestions for Authors
The manuscript reports a study exploring the spatial scaling behaviors of space-based CO2 measurements and their spatial distribution in the regions of China. A multi-fractal detrended fluctuation analysis (MDFA) was applied in the study. The data were from the GOSAT satellite, and missing data were interpolated using a spine interpolation method. The study confirmed the XCO2’s multi-fractal behaviors and discussed influencing factors. It is interesting to see how different the behaviors are between the regions of China.
Overall, the manuscript is well-developed, as I can follow the narratives and complex formulas with my limited knowledge of MEFA. It will be helpful if terms like “long-range correction” and “a broad density function” can be explained with physical meanings and discuss what they imply to CO2 emission and ecosystem CO2 uptake behaviors in each region?
We appreciate your comments, suggestions and advices. Your comments are really very marvelous and give us a great help to improve manuscript quality.
The term “long-range correction” is a spelling mistake. The term “long-range correlation” is right to depict auto-correlation. Suppose that there is a time series {x(t), t=1,2,3,,,N} with length of N,
Rhu(tao)= <x(t)x(t+tao)>/tao
is defined auto-correlation function. Thus when auto-correlation is persistent, x(t) is correlated with x(t+tao).
The term “a broad density function” is omitted a word “probability” in our paper. The term “a broad probability density function” is right.
Kantelhardt et al.
(Kantelhardt, J.W.; Zschiegner, S.A.; Koscienlny-Bunde, E.; Bunde, A.; Havlin, S.; Stanley, H.E. Multi-fractal detrended fluctuation analysis of non-stationary time series. Phys A. 2002, 316, 87–114. )
pointed out there are two types of multi-fractality sources: one is a long-range correlation and the other is a broad probability density function. Two kinds of physical mechanism are different, the former meaning is that x(t) is correlated with x(t+tao); and the latter meaning is that x(t) is uncorrelated with x(t+tao), when x is small, its probability density is possibly high; oppositely when x is great, its probability density is likely to be low. A probability density function is asymmetry.
Global warm is a challenge of mankind. To overcome this challenge, Co2 neutralization is an important and urgent task for every country government. Variation in Co2 concentration is affected by many factors. Their physical mechanisms are different. With respect to the cases of long-range correlation, we can properly relax restrictions on carbon emission in some regions with high land-cover of forestry owing to high absorption; but in other regions with low land-cover forestry, we should strictly restrict carbon emission and more vegetation plant is encouraged to absorb CO2 in the atmosphere. Regarding region with a broad probability density function, we should reduce abrupt events of Co2 emission in order to change a broad probability density distribution.
Hence, this investigation attempt to provide a theoretical support for policymaker and to predict Co2 concentration regarding vegetation absorption and climatic change.
Just a few places need revisiting.
Line 7-8, the first sentence of the Abstract sounds unnecessary.
We have deleted them
Line 91, delete “Fig. 1 shows”
We have deleted them
Line 101, the figure caption needs to offer more information about each study area so that the readers can know about the land cover/vegetation/climate A, B, C, and D without referring back to the text. Also, in the text, explain why or how to choose these four locations? Please apply this comment to other figures’ captions.
We have added Caption for Fig.2, Fig.5, and Fig.6.
The four typical grid points are depicted in section 2.1
Line 128-208, double-check all equations and related variable symbols and explanations
We have checked them
Line 145, any citation for the STPS method?
STPS is referred to spatial thin plate spline interpolation method. In order to avoid ambiguity, Line 145 is corrected. See 154 in a reviewed version.
Line 340-341, provide more explanations of “long-range correction” and “a broad density function” before these terms are used.
We have discussed in the above.

Reviewer 2 Report
The article has certain research value and significance. Please make the following modifications before receiving the article:
1、 Please provide the drawing software in this paper.
2、 Please delete lines 86-88. It's not necessary.
3、 Please beautify Figure 2. The ordinate unit and legend cannot be seen clearly.
4、 North compass is missing in the map, please add.
5、 Please supplement the meaning of the subgraphs in Figure 5 below the drawing name.
6、 The discussion part is too simple. Please strengthen the depth of the discussion.
7、 The conclusion part does not need a conclusion paragraph, but can be listed in the form of the first, second and third.
Author Response
Review#2
Comments and Suggestions for Authors
The article has certain research value and significance. Please make the following modifications before receiving the article:
We appreciate your comments, suggestions and advices. Your comments are really very marvelous and give us a great help to improve manuscript quality. Thank you for your comments and suggestion.
1、 Please provide the drawing software in this paper.
In the beginning, spatial coordinate is visualized into map by ArcGIS 10.5(Www.esri.com), next, ArcGIS 10.5 is used to export vector’ map into JPEG format map. In the end, the maps with JPEG format are edited in the software Photoshop (WWW.adobe.com). Because licenses of these software are administrated by our university, and these licenses are solely used to educate and train university graduate students. “Please provide the drawing software in this paper”, I think hard, but do not understand your concerns very much. We have applied to our university. If they agree, we can.
2、 Please delete lines 86-88. It's not necessary.
We have deleted them.
3、 Please beautify Figure 2. The ordinate unit and legend cannot be seen clearly.
We have beautified them.
4、 North compass is missing in the map, please add.
We have added North compasses.
5、 Please supplement the meaning of the subgraphs in Figure 5 below the drawing name.
We have added caption.
6、 The discussion part is too simple. Please strengthen the depth of the discussion.
CO2 concentration in the atmosphere is closely connected with carbon emission/absorption. Clearly, carbon emission can increase CO2 concentration, but vegetation absorption, soil absorption or sea absorption can deduce CO2 concentration. Moreover, climatic changes have great effects on CO2 concentration. China covers a large geographical area, with wide climatic variety, complicate topography and plentiful vegetation coverage. Within this context, it would be interesting and ideal to examine the spatial distribution of the multi-fractal properties over the whole of China. Spatial distribution of multi-fractality of CO2 concentration is influence by air temperature and vegetation growth.
We speculated that small fluctuation and large fluctuations of CO2 concentration have influence on multi-fractality of CO2 concentration, but about vegetation growth, small fluctuations of vegetation absorption Co2 seems multi-fractality of CO2 concentration. Due to absent data, further investigation is needed in order to acquire a more reliable results.
We have strengthened the depth of the discussion. See Section 5.
7、 The conclusion part does not need a conclusion paragraph, but can be listed in the form of the first, second and third.
We have corrected them.

Round 2
Reviewer 2 Report
I think it can be accepted now.